# Helping Women Suffering from Drug Addiction: Needs, Barriers, and Challenges

**DOI:** 10.3390/ijerph192114039

**Published:** 2022-10-28

**Authors:** Marek A. Motyka, Ahmed Al-Imam, Aneta Haligowska, Michał Michalak

**Affiliations:** 1Institute of Sociological Sciences, University of Rzeszow, 35-959 Rzeszów, Poland; 2Department of Computer Science and Statistics, Doctoral School, Poznan University of Medical Sciences, 61-806 Poznan, Poland; 3Department of Anatomy and Cellular Biology, College of Medicine, University of Baghdad, Baghdad 10001, Iraq; 4Barts and the London School of Medicine and Dentistry, Queen Mary University of London, London E1 2AD, UK; 5Doctoral School, University of Rzeszów, 35-959 Rzeszów, Poland; 6Department of Computer Science and Statistics, Poznan University of Medical Sciences, Rokietnicka 7 St. (1st Floor), 61-806 Poznan, Poland

**Keywords:** social stigma, women, drug use, addiction treatment

## Abstract

Statistical data on the use of various psychoactive substances indicate a narrowing of previous differences in substance use between men and women. Data from studies conducted among women suffering from drug addiction are increasingly published, with the authors highlighting the specific needs of this group and the difficulties that women with addiction problems encounter. The current study aimed to identify the barriers and needs of this audience, both when seeking help and during treatment. The method used in the study was secondary content analysis. To identify publications describing the barriers and needs of women suffering from drug addiction, we searched the PubMed database to find publications that met the adopted research objective. We set the data search period to the last ten years to examine the timeliness of the issue under study. The search yielded 199 research reports. Twenty-three articles describing 21 studies were included in the final analysis. The selected publications dealt with the difficulties and challenges faced by women with addiction problems. Barriers to accessing treatment for this group, the needs, and the challenges of helping women suffering from addiction were identified. Results showed that the barriers are mainly stigma but also deficits in the therapeutic offerings for this group. The primary need was identified as the introduction of appropriate drug policies, and the challenges, unfortunately, are the still-reported gender inequalities. To improve the situation of women, regular attention to these issues and the need to include them in national health strategies is essential.

## 1. Introduction

Indications of psychoactive substance use between men and women are increasingly equalizing, and helping women suffering from substance abuse is a significant challenge for today’s social and treatment services. In the 1980s, the estimated male-to-female ratio for alcohol abuse was much higher, at 5:1 [1]. However, in the first decade of the 21st century, statistics on the use of various psychoactive substances indicated a narrowing of the earlier gender gap [2]. The World Drug Report 2021 estimated that one in three users of psychoactive substances is female; the data confirm that the percentage of indications between men and women is becoming more similar, both in terms of the use of these drugs and reporting to drug treatment facilities [3,4]. Estimates indicate that the percentage of women using these substances ranges from 10% in Asian countries to 40% in European countries [5]. Data also confirm that deaths among female drug users are increasing [6].

Only a tiny percentage of women and men who try any psychoactive substance will develop an addiction [7], but the problems experienced by women drug abusers are a consequence of fundamental differences in this group compared to men. These relate to the reasons for reaching for drugs, the greater vulnerability to developing an addiction, the complex difficulties of recovery, and the greater risk of relapse. Already young girls have many more risk factors than boys, and they have more extensive problems in various aspects of life, which increases the risk of serious drug use problems in adulthood [8]. Research confirms that alcohol or drug initiation stems from the need to cope with anxiety, low self-esteem, depression, feelings of isolation, and the trauma of sexual abuse and violence among many adolescent girls and women. For men, on the other hand, initiation is often driven by the need to belong to a group [7,9,10].

Women are more likely than men to consume alcohol in response to stress, may be particularly susceptible to the reinforcing effects of stimulant drugs, are more likely than men to engage in non-medical use of prescription opioids, and are more likely than men to use additional drugs (e.g., sedatives) to enhance the effects of opioids. Although data on the differences in smoking and marijuana use indicate higher prevalence among men, women are estimated to have more difficulty quitting, while marijuana use can cause severe premenstrual syndrome or premenstrual dysphoric disorder. In addition, ninety percent of cases of anorexia nervosa and bulimia nervosa involve women, and data on gender differences in behavioral disorders estimate a 2–3 times higher prevalence in women [2]. These differences are likely the result of gender-specific biological mechanisms interacting with sociocultural influences and life stressors that affect women and men differently [9]. Data from some studies underscore the importance of sex hormones in modulating drug effects in women [11]. In addition, women transition from substance abuse to addiction much more frequently and quickly than men; they are more likely to self-medicate with illicit substances; they are more susceptible to the health consequences of substance abuse, overdosing, and they experience relapse more often than men [7,12]. 

Observers and researchers of women’s drug use point to the specific characteristics of this group, but especially to the difficulties women encounter when deciding to seek treatment. The image established in the social space of men with an addiction problem differs significantly from that of women suffering from addiction, who face ostracism and stigmatization during the development of addiction and during treatment attempts [13,14,15,16]. Negative attitudes toward women experiencing addiction are related to the consequences attributed to women who use these substances, including unwanted pregnancy, prostitution, and transmission of infectious diseases [5]. In addition, women who are mothers suffering from addiction have a complicated situation stemming from the stigma they experience compared to non-mothers and the need to balance parental responsibilities with participation in therapy, often lasting several weeks in an inpatient setting [17].

## 2. Purpose

Our study aimed to identify and analyze the most frequently indicated difficulties encountered by women who abuse psychoactive substances (other than alcohol) due to sociocultural conditions and other difficulties experienced when trying to undertake treatment, during, and after treatment. We chose difficulties related to the abuse of psychoactive substances other than alcohol because problems related to alcohol use that researchers more frequently described in the scientific literature; in addition, the use of psychoactive substances other than alcohol is met with intense social stigma, especially against women [17]. Although the results of quantitative analyses do not seem to support this, qualitative studies confirm the strong stigma experienced by women suffering from drug use [18].

## 3. Material and Methods

Secondary analysis of empirical data, in this case, articles collected from a database of online scientific publications, was adopted as the current research method. We adopted three indicators of the difficulties faced by women suffering from drug addiction: (1) barriers to attempting help and change; (2) needs, especially those related to therapy; and (3) challenges, the reduction of which may require substantial changes in sociocultural scope.

Research questions were prepared and answered in selected publications to identify the indicators adopted:What barriers to entering treatment are reported by women suffering from drug addiction?What needs of women struggling with addiction have been identified in the selected studies?What challenges do policymakers face that, if reduced, could contribute significantly to improving the situation of women suffering from drug abuse at the help-seeking stage, during, and after treatment?

The indicators and formulated research questions determined the scope of our research efforts. A comprehensive search of the PubMed international database of peer-reviewed scientific publications was undertaken to find articles that described research conducted in various parts of the world with women suffering from drug abuse. To check the timeliness of the indicated problems, we set the time frame of the searched publications to the last ten years while searching for English-language publications only. The selection of articles was made by searching only those that dealt with studies conducted among women drug users, excluding review publications that also referred to studies conducted in different time frames, even in the second half of the 20th century. In our selections, we considered only those studies in which ethical considerations for conducting research with human subjects were considered. We entered keywords into the search engine at https://pubmed.ncbi.nlm.nih.gov/ (accessed on 17 July 2022) that were relevant to our search and could answer the research questions we formulated: “women” and “drug users.” We identified 8989 articles in the PubMed online database that contained the indicated keywords. After supplementing the previous keywords in the search engine with three more: “needs”, “challenges”, and “barriers,” 199 publications were found. Most of them, however, were excluded because they involved many studies in which, although the keywords were present, the articles addressed a different issue. After a detailed review by two authors (M.A.M and A.H.), 23 publications describing 21 research initiatives that were relevant to our work were selected. During the selection process, it was found that one study was described in three articles, but these publications described the research conducted in different contexts; the first on the sexual and reproductive health needs of women in Kenya [19], and the second dealt with the perception of stigma experienced by the research participants [20], and the third described the needs identified in the sample included in the study [21]. It was decided to include all three texts because of their valuable data. 

The articles were selected in terms of their descriptions of women’s problems; that is, the publications had to fit within the predetermined inclusion criteria strictly and describe actual research on the needs of women suffering from drug abuse, but also on the barriers that make it difficult for them to get treatment and change their difficult situation. At the same time, during the selection process, it was noted that some identified barriers and needs could be considered challenges, which are much more difficult to reduce than the searched needs and barriers. In our understanding, challenges are phenomena of much greater scope than just economic or systemic barriers. Challenges are cultural phenomena that require major transformations to change, which are, unfortunately, complicated to implement in many countries.

Analyzing empirical material collected from only one database can be a limitation. However, this was done consciously, knowing that the PubMed database is one of the databases containing high-quality peer-reviewed publications. In addition, the choice of PubMed as the primary database to look for relevant data was influenced by the fact that this database provides free access to scientific publications found in MEDLINE and some articles from other scientific journals. The MEDLINE database includes publications from more than 5200 journals from more than 80 countries. In our opinion, it is a rich data source conducive to the reliable exploration of scientific materials [22]. The authors do not rule out conducting additional similar studies in other databases, e.g., to check for differences in the number and value of publications on similar topics; this, however, may be an idea for a separate study.

## 4. Results

The 23 publications selected were the result of research undertaken in several countries around the world, both in Europe, North America, Asia, and Africa (Canada, Iraq, Tanzania, Georgia, USA, England, Vietnam, Thailand, India, Kenya, Iran, Spain, Indonesia), which provides a relevant and valuable insight into the issue addressed, and also allows a comparison of the difficulties faced by women suffering from addiction, especially in terms of cultural differences. Despite the diverse regions of the world and cultural backgrounds, the publications identified many circumstances in common, which allowed them to characterize the most significant barriers, needs, and challenges for women struggling with problems resulting from drug abuse. We included a detailed description regarding the selected publications (authors, year of publication, scope of study, sample size, and results) in the table in the Appendix A.

### 4.1. Barriers

Reported difficulties in accessing treatment vary, but in most of the studies analyzed, women suffering from drug abuse point to stigma as one of the main barriers keeping them from seeking help or entering treatment. In the United States [17,23,24,25] and Canada [26,27], where drug treatment appears to be at high standards, stigma is still a significant and ongoing problem. The same is true in European countries, such as Spain [28] and Georgia, where researchers report extreme forms of stigmatization in the form of social hostility toward women suffering from drug use, intolerance on the part of their families, and lack of support from them, and even repression of users and, in the case of necessary hospitalization, being subjected to critical judgments by medical personnel [29]. Analogous difficulties were reported in studies conducted in Asian countries, such as Vietnam [30], India [31], Iran [5], and Indonesia [32]. 

Another frequently reported barrier is the lack of therapeutic offerings and programs specifically tailored to the unique needs of women [28,29,32], the low effectiveness of proposed interventions [17], the lack of understanding of the needs of this group by providers [20], the lack of special rehabilitation centers for women [5], and, in the case of women with addiction problems serving prison sentences, the lack of access to adequate health services [21]. In addition, difficulties in accessing services often associated with drug abuse are reported: a lack of interventions tailored to deal with violence for women who use drugs and a lack of services focused on women’s needs, particularly sexual health, reproductive health, and childcare [28]. 

However, if offers of help are available, economic disadvantages are a barrier for women suffering from drug abuse: sometimes extreme poverty [26], and often simply unfavorable socioeconomic conditions that preclude participation in paid treatment programs [5,20,23,29,31,33,34,35]. Barriers also include a lack of transportation, difficulty in accessing social services and health care [23], limited access to diagnosis, treatment, and successful health outcomes [28,32,36], as well as long distances to health care providers, lack of communication with health care providers [20], and sometimes lack awareness of existing social support services [5], or lack of awareness of one’s own health needs, with a subjective lack of need for health services [24].

Significant barriers that keep women drug abusers from seeking treatment are their emotionality, or rather, their problems with coping with unpleasant feelings, such as shame and helplessness [25,26,27,31], guilt, and lowered self-esteem [29,37], feelings of loneliness [33], severe anxiety, depressive states, permanent stress and other emotional disorders [37,38,39], and, above all, low competence in coping with these unpleasant states and the abuse of drugs to suppress these states and the subjective sense of benefit resulting from their reduction [17,37]. 

Difficulties identified in the studies analyzed, which inhibit the process of changing the current lifestyle, are, in addition, the need to conceal problems related to drug use to obtain complete or better health care in medical facilities [26], reluctance to disclose one’s problems [37], low acceptance of addiction as a disease and lack of knowledge about the dangers of using psychoactive drugs [32,39], racial and gender-based violence [27,28], marginalization and discrimination [19,23,25,27,39], including repression of drug users [29]. In addition, studies among Vietnamese women have identified low access to HIV testing and a lack of knowledge about sexually transmitted diseases (STDs) and viral hepatitis [30], and among Kenyan women, low accessibility to sexual health services [21], as well as stigmatization of HIV-positive women by other drug users [19].

### 4.2. Needs

The barriers identified in the studies point to needs that should be realized to reduce drug use and make it easier for women suffering from addiction to decide on treatment, participate fully in the treatment process, and provide adequate support once completed.

The crucial interactions indicated in the analyzed publications are, first of all, appropriate drug policies, and within this framework, especially the introduction of preventive interventions to prevent drug initiations [24,31,32,33,36] as well as harm reduction interventions for women who have already had their first contacts with drugs [19,20,21,24,31,33,36,38]. Many researchers indicate that these interactions are essential for improving education and awareness of risks. One of the most critical needs, however, seems to be the establishment or increase of facilities exclusively for women with addiction problems, as well as medical facilities for pregnant women or women with children, and above all, the adaptation of treatment programs to the specific needs of women [26,27,28,29,30,35,39]. It is also important to introduce appropriate social, health, and care interventions for women suffering from drug addiction before and after drug treatment [5,23], if possible, with the introduction of interventions implemented by women for women [32]. Of course, for these interventions to be implemented, adequate preparation of medical personnel is necessary, especially therapists’ high cultural competence, empathy, unconditional positive regard, and authenticity (Rogers’ triad) [25]. The sensitivity of staff when working with women with drug use problems [40], the education of medical services on the needs of female substance users [29], and therapeutic work with a patient with a possible history of sexual assault and at risk of pregnancy and past or present sexually transmitted diseases [37] seem particularly valuable. 

It is, therefore, vital to provide training for healthcare professionals to become competent in providing services to women who abuse drugs [21], as well as in respecting human rights and medical ethics [28]. Unfortunately, some authors report a lack of adequate funding to meet the indicated needs [21,36,39]. 

Needs related to therapeutic offerings include modification of existing therapy programs and introduction of interventions for coping with stress, anxiety, low sense of value and efficacy, shame, reduction of depression [19,34,40], supplementation of knowledge on procreation, and possibilities to protect against unplanned pregnancy [21,34], education on HIV transmission [30], education on alternatives to drugs for coping with unpleasant emotions and training in social competence [5,17]. It is also essential to introduce sessions within family therapy that allow women to be understood and accepted after discharge from rehabilitation centers [5], and to educate women about the rights they have, especially in African countries where male dominance seems particularly strong [19,20] and to cater to services for women over the age of 40 years [26].

Indicated needs of a social nature include broad public education about addiction as a medical condition, especially in countries with extreme stigmatization of sufferers [29]; improving the safety of community interventions in low-income housing [27]; activating women substance users to stay in touch with medical and social services [37]; greater social integration with people outside the substance abuse community that can foster the building of a support network for undertaking lifestyle changes [17]; support from the family, especially the husband or partner [32]; and improving living conditions [27]. 

Academics are also addressing needs. Slightly more than two dozen studies in a decade indicating the needs of women with addiction problems and the barriers to implementing change seem insufficient for decision-makers to decide on appropriate interventions. That is why researchers are primarily signaling the need for all scientific initiatives to identify the problems of women drug users [38], especially in the area of determining predictors of drug use among women [33], and, most importantly, holistically determining the health needs of women with addiction problems [23].

### 4.3. Challenges

These are perhaps the most complex demands for change, driven by both barriers and needs but may be particularly difficult to implement, especially in countries where women’s social position appears to be culturally positioned much lower than that of men. 

The most important aspects seem to be drug policy changes, especially in Georgia, where current policy in this area–as the study authors suggest–maintains the sociocultural conditioning of negative attitudes toward drug users-especially toward women. The extreme marginalization of this group is both a barrier for women suffering from drug abuse to make any attempt at treatment and a severe challenge for policymakers [29]. The same is true, by the way, in African countries, including Tanzania, where fear of social consequences, violence against women, and low access to prevention and treatment services seem to be more of a challenge than a need [38], especially since research and prevention initiatives seem to be rare in that country [41]. In addition, drug abuse and HIV risks are a “pressing problem” that should not be ignored [42]. Another African country where helping women who abuse drugs seems to be a challenge for policymakers is Kenya. Here, the authors of the research analyzed point to the need to change social norms in the perception of women [21], without which proper assistance is hardly possible, as the existing perception of this disadvantaged group promotes isolation, exclusion, discrimination, and stigma in health care facilities [19]. Without the introduction of an appropriate drug policy, the challenge seems to be to sensitize providers to the needs of women who inject drugs, to build an atmosphere of trust conducive to disclosing their problems [20], and especially to integrate drug treatment, family planning education and sexual health services with other community services [21], resulting in, among other things, a reduction in HIV infection and improvements in women’s mental and somatic health.

Introducing appropriate policies in other countries also seems challenging, especially in Asian countries, where a strong stigma against women drug users is a barrier to seeking help. Challenges include introducing sociocultural changes directly or indirectly to improve the situation of women suffering from addiction, as well as modifying existing drug policies, e.g., in Iran, creating special rehabilitation centers for women to help them meet their specific needs, introducing free drug treatment, reducing family disputes [5], in Indonesia, introducing treatment policies and services that are sensitive to women’s experiences and take into account their gender and needs [32]. Education of women, who often plunge into drug use problems due to lack of knowledge, is also challenging. A challenge in India is the meager education of women, the lack of knowledge about the consequences of drug use and sexual health, and the low need for healthcare services [31]. In Vietnam, where marginalization of this group is a frequently reported barrier, challenges include intensifying blood-borne infection prevention programs, psychological support for women, counseling, education on family planning and parenting skills, maintaining or improving the identity of a good mother, and reducing fear of their children’s future [34]. Challenges reported by Iraqi researchers stem from the needs identified; these are primarily the introduction of prevention education to prevent early drug initiation, the high risk of health complications associated with the simultaneous use of sedative drugs and stimulants, and the introduction of interventions to reduce risky sexual behavior associated with the need to obtain funds for drugs [33]. It should be noted that other studies conducted in Iraq have not found a high prevalence of such problems [43].

Challenges necessary for change are also noted in European countries. Publications analyzed indicate that in Spain, challenges include structural barriers to accessing harm reduction services, including limited resources that can improve the situation of women in a drug abuse crisis, stigma and discrimination from healthcare professionals, denial of care, provision of substandard care, physical and verbal abuse, more extended waiting periods for women, transfer of care to younger colleagues, disclosure and confidentiality issues, and personal moral judgment of women with substance abuse problems [28]. In England, on the other hand, structural challenges include poor access to doctors’ offices and counseling related to both treatments of substance abuse problems and sexual health issues, as well as access to related institutions, e.g., related to anti-violence and sexual assault [37]. In addition, in our opinion, impoverished social relationships that increase the reluctance of women in need of help to meet with specialists, as well as self-protective measures against the loss of “good” well-being, and especially social stigma that unfortunately effectively prevents decision-makers from seeking help and providing it at the same time, can also be a challenge [40]. 

The challenges identified in the countries where egalitarianism in terms of access to treatment and equal rights for women struggling with drug abuse problems seem to be both relevant and, unfortunately, a much-neglected social norm. The authors of the publications analyzed highlight some complex challenges. These include improving social factors such as women’s housing stability and personal safety; health systems creating safe spaces that allow women with limited social stability to access care free of charge [35]; establishing women-appropriate centers with adequate therapy and well-trained treatment staff; and integrating social environment interactions into prevention, intervention, and treatment [17]. As it turns out, the challenges are still education about health needs, both in terms of medical help for this group and in terms of introducing appropriate preventive interventions, bringing in empathetic, non-stigmatizing providers willing to fully commit to helping this group [24]. At the same time, therapists’ high level of training and egalitarianism are advocated, especially by African-American women suffering from addiction [25]. For women with addiction problems serving prison sentences, challenges related to their lowered self-esteem and sometimes harsh economic conditions include avoidance of seeking help in public institutions (hospitals, schools) due to fears of mistreatment for having been incarcerated in the past, and poor mental health as a consequence of their overall life experiences [41]. In addition, for women living in poorer neighborhoods, an indicated challenge is the so-called “domino effect,” i.e., despite the favorable completion of therapy, recurring difficulties in returning to fulfilling social roles related to past experiences [23]. In contrast, for women living in remote, peripheral rural areas, challenges include marketing efforts to distribute painkillers that encourage the use of such drugs, including strong painkillers, the unproblematic prescribing of potent painkillers by rural doctors, and at the same time, limited treatment options [36]. 

Researchers from Canada emphasize that gender inequality is a significant challenge in their country: the culturally perpetuated image of women as docile, subservient, and subordinate to men, the subjective treatment of women who abuse substances [27], perpetuating discrimination, resulting in a hierarchy, i.e., exploitation of the position: staff-woman with addiction, as well as ageism; difficulties in accessing specialized help for women as young as 40 years old [26].

## 5. Discussion

Many publications indicated that stigma toward women suffering from drug addiction is the most frequently reported problem. Social beliefs expect women to be home caretakers, raise children, and be more family-oriented than men. Women are usually aware of these double standards and hide their drug use [17]. Women who identify an addiction problem in themselves and consider seeking treatment often do not seek treatment precisely for fear of being stigmatized or, if they are mothers, for fear of being restricted or losing parental rights [44,45]. It also happens that even the closest family members are opposed to them when seeking help, not believing in their powerlessness to undertake sustained abstinence on their own [45]. Some women feel guilty about their loved ones and try to reduce their guilt by ignoring and hiding their substance use and rationalizing their behavior by claiming that the pleasurable effects of drugs are more attractive and more important than a drug-free life [46]. 

In addition, support services and other social workers make women entering addiction treatment more likely to be stigmatized than men. In addition, they have lowered self-esteem and find it difficult to maintain supportive relationships due to their previous experience with violence [47]. The attitudes and beliefs of healthcare professionals significantly impact the treatment of addicts by shaping treatment practices and research priorities. However, many health professionals prefer not to work with addicts [48], which is often associated with what is known as contagion stigma, which means that those who research or treat highly stigmatized populations may also experience stigma [49]. Stigma also shapes treatment practices, and as many researchers note, women are not studied to the same extent as men in clinical and preclinical research [50,51]. The current focus on women’s health and gender differences has brought public attention to the interpretation of these differences in therapeutic and diagnostic decision-making. This approach today stems from strong feminist movements [45].

The social status and economic situation of addicted women are much worse than that of addicted men. They are much more likely to be unemployed or have lower incomes while working, which becomes double jeopardy in the case of motherhood [52,53,54]. For women seeking help and entering therapy, recovery and regaining proper social roles (mother, partner, wife, employee, and others) is often complicated. Their resources (family, housing, health, education, work, and friends) are often insufficient to help them consolidate abstinence. Many are homeless and have nowhere to return to, especially if they have served prison sentences for drug-related crimes [2,55,56]. Their families of origin are often families with addiction problems, or family relationships have been so damaged that they can hardly rely on relatives for support [57,58,59], and they have low educational attainment [59]. Further down the line, they have never worked or have been unemployed for a long time [56], feel lonely [60,61,62], and experience severe health damage (both somatic and mental) related to psychoactive substance use [44,63]. These problems are multidimensional, intricate, and notably acutely experienced.

Many women who use substances may have mental health problems (depression, post-traumatic stress disorder, eating disorders, or paroxysmal anxiety); according to statistics, women addicts often contemplate suicide or are victims of suicidal actions [2,64]. Mental illnesses, phobias, or depression can promote the abuse of mood-altering psychoactive drugs; there is a high degree of co-occurrence of drug abuse with psychiatric and mood disorders [44]. Evidence suggests that adolescent psychoactive substance use can cause psychotic symptoms and mental disorders in later years [65,66,67]. It should also be added that somatic illnesses and various emotional situations (falling in love, losing money, taking away children, divorce) can be inflammatory factors for using psychoactive substances or symptoms of addiction relapse [68]. 

Another problem is low self-esteem, which can stimulate the mechanism of initiation and substance abuse and cause addiction and relapse. However, low self-esteem alone does not explain the mechanism of substance use. Attention should be paid to the individual’s social environment, which significantly influences the behavior in question [68]. Many researchers believe that psychological distress and low self-esteem influence the use of psychoactive substances [69]. Women who use these substances, in addition to temporarily raising their self-esteem, “enter” their social reality in which they feel good. By using drugs, they find a sense of self-esteem, strive to perform tasks, and achieve goals they believe society sets for them. For them, drug use is a way to cope with feelings of stigma or disrespect for a short time, superficially boosting their self-esteem [17]. For many, feelings of guilt and shame over inappropriate relationships with others, especially over inadequate care for their children, represent the most challenging area of self-forgiveness [70]. In addition, evaluative and critical societal views about the prevalence of addiction in women–even after a happily completed treatment program–reinforce their feelings of powerlessness. During treatment, the mere obligation to continue the process by attending support groups can increase their feelings of inadequacy, marginalization, and stigmatization [17]. Therefore, it is crucial to tailor the treatment process to women’s specific problems.

The problems of women in addiction crisis have been more widely and frequently publicized for several decades, but deficits in the support system for this group continue to be reported. For women who inject drugs, underfunded healthcare systems and barriers to accessing harm reduction and addiction treatment programs are significant obstacles to accessing care and entering treatment [71]. Women suffering from addiction who undergo the therapeutic process in coeducational wards find it challenging to properly participate in the treatment process. These difficulties are due to both the differences in the dynamics of addiction in this group (causes, pace, consequences, emotions) and the insufficient competence of therapists, who, for various reasons, overlook the specificity of the problems present in women when implementing therapeutic programs or deal with them in a defective manner [57]. 

A critical factor discouraging women from entering therapy is childcare during therapy. For many women, losing custody of their children is a significant threat and a barrier to treatment, although, for a significant proportion of participants, it is sometimes a motivation to seek help. In addition, women report social stigma as a barrier to treatment in personal and professional contexts [46]. Women who seek treatment pointed to the limitations of gender-sensitive treatment programs that may not consider their increased domestic responsibilities. Sometimes, with no choice, many continue to use psychoactive substances [72,73]. 

Among the significant barriers to treatment cited is the poor and unstable economic situation of women related to their past drug use, and the increasing financial problems, making it difficult to access medical services and addiction treatment. In addition, women needing support for their plight hint that it is often difficult to find transportation to treatment facilities because they cannot afford it or do not have other support from which to draw [46]. 

The factors we mentioned earlier play an essential role in seeking help in the treatment and maintaining sustained abstinence afterward. It is worth noting that only an adequately integrated model of care (e.g., case management or sharing mothers with their children during therapy) can make a significant contribution to improving women’s functioning and, at the same time, reducing the number of crimes committed and returning to fulfilling the social roles and tasks assigned to these roles; mother, worker, citizen, among others [68]. 

However, good practices conducive to improving the situation of women suffering from addiction are indicated. Ayon et al. (2019) showed that reaching out to the community is a feasible model through which all relevant services can be introduced, including family planning, sexual health, social support, preventive interventions, and others [21]. This publication initially outlines the needs of female drug users and stakeholders who can assist and then presents designed interactions to meet those needs. The current study highlights the value of staff training, human resource capacity building, technical support, and financial resources to provide contraception and other sexual health services. A developed and integrated assistance system that pays attention to and targets women’s concerns has tangible benefits. Valuable findings include indications that strengthening organizational capacity and human resources, adequate technical support, access to financial resources, and public acceptance of these needs and changes in perceptions of women are critical factors for change [21].

Also, according to Akré et al. (2021), addressing the social needs (such as food, clothing, safety, and housing) of women who use drugs can support better access to healthcare services. Therefore, these are further important indications for policymakers responsible for accessing and developing medical and social care for women suffering from substance abuse [35].

The demands developed in 2019 for intensifying and improving support for women with substance abuse problems are, first and foremost: studying the impact of substance use on different subpopulations of women (including: LGBTQ+ or cultural minority women); identifying the specific needs of women of reproductive age, particularly treating substance use disorders in pregnant or post-pregnant women, organizing better access to health and family planning services; developing appropriate treatment strategies for women with substance abuse problems that take into account experiencing the trauma of violence or ostracism due to substance use; developing treatment strategies for women who are parents or caregivers; developing support strategies for women that include those closest to them; including loved ones in the treatment process to better understand the suffering and difficulties of women experiencing addiction; developing strategies to help balance cultural differences in perceptions of women with addiction problems; and using mobile health (m-health) technology to develop flexible information, screening and treatment services for women experiencing addiction [74]. The solution, moreover, could be the development of women’s services for women or the creation and development of treatment facilities serving only one gender.

## 6. Conclusions

The difficulties and barriers outlined highlight existing support options and point to the need to review access to treatment for this group; this is especially true for young women, starting early in their substance abuse journey, unaware of their entry into dangerous activity, and above all cut off from access to adequate support. Furthermore, this is not due to a lack of institutions that can provide such assistance, but sometimes due to a lack of adequate knowledge, lack of access to institutions, lack of adequate response from aid institutions, or reprehensible behavior by public officials, and sometimes due to stigma or a high fear of experiencing it.

The postulated interactions are the introduction of appropriate public education, but above all, the proper preparation of the healthcare system to meet the needs of this group, the introduction of preventive measures aimed at girls in adolescence, as well as women leaving addiction treatment facilities or prisons where they were incarcerated due to crimes committed under the influence of substances. The former group requires changes in women’s sociocultural patterns and, especially, in the roles attributed to women, but in this case, the change process appears to be more complex than simply introducing appropriate health policies.

An action we recommend in our deliberations is also to use methods of artificial intelligence and machine learning to even more fully reach publications that take into account the topics covered, allowing monitoring and especially the early response to the needs and challenges established in the latest research for this audience [75].

Paying attention to this issue and the need to include it in national health strategies seems essential for improving the situation of women struggling with addiction; it can influence the introduction of appropriate measures to encourage women struggling with addiction to enter treatment while facilitating their return to the proper performance of all social roles and tasks associated with them. Addressing this issue is warranted because gender differences in substance use disorders are still poorly studied.

## 7. Limitations

A limitation of the current study is the search for materials for analysis in only one database of scientific articles. However, this was dictated by the desire to check the resources available in PubMed, considered one of the most valuable databases. Comparing the data obtained with resources from other databases may be of interest for a separate article on a similar topic.

## Data Availability

Not applicable.

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
