# Peer review of "Helping Women Suffering from Drug Addiction: Needs, Barriers, and Challenges"

_ijerph, 2022, doi:10.3390/ijerph192114039_

Round 1

Reviewer 1 Report (New Reviewer)

Please see attached

Author Response

Thank you very much for evaluating our work and for all the comments to which we responded by making corrections and additions to the text. Unfortunately, in one case we were unable to respond (comment to page 3 line 125). We will admit that this is difficult for us to explain. This is a very valuable comment, which we will certainly use in the future. We hope that it will not influence the decision not to allow publication.

Reviewer 2 Report (New Reviewer)

This is a clear and well-written article that makes an important contribution regarding women with substance use disorder. It is especially noteworthy that the authors have examined research from a number of different cultures--this is a critical strength of the work.

The article illustrates how many important ideas and recommendations can be gleaned from a rather simple literature review. Another critical strength of this work is that recommendations to improve the lives of women abound! Recommendations to improve the lives of women range from increasing access to treatment all the way through policy changes, and everything in between. The authors did a good job of explaining how and why they developed the categories of barriers, needs, and challenges.

I have only a very few suggestions:

Consider amending the title to remove "in search of" so the title can read: "Helping women suffering from drug addiction: needs, barriers and challenges"

The introduction/literature review would benefit from the inclusion of material on women as mothers. Yes, there are comments about motherhood later in the article but this deserves to be front and center as a critical issue for women. Mothers endure greater stigma, have fewer treatment options, and bear the brunt of locating childcare in order to receive treatment.

Regarding the abstract, it can be simplified and read a bit better by removing the words Introduction, Purpose, Results, and Conclusions.

Overall, a very good effort!

Author Response

The authors thank you for your very favorable review of our work.

Thank you very much for your valuable comments, according to which:

  1. we have changed the title;
  2. we added a paragraph about mothers;
  3. we removed unnecessary words from the abstract.

Reviewer 3 Report (New Reviewer)

The text is highly interesting and informative, and the results of the analyses have practical applications. The following comment in no way diminishes my high assessment of the merits of the article:

It may be worth mentioning (which I leave solely to the Authors’ opinion), whether, for some reasons should it not be possible to create a therapy group exclusively for addicted women, selective separate therapeutic sessions only for women would be helpful while working with heterogenous groups.

The above comment is merely a suggestion and does not impact my recommendation for the text to be printed in the as-is form.

Author Response

Thank you very much for such high recognition of the quality of our work. Thank you for your comments regarding the possibility of creating therapy groups for women. Content on this topic has been included in the text.

Reviewer 4 Report (New Reviewer)

It is a very well-written article, the subject of which shows not only the problems with the therapy of drug-addicted women, but also shows the problems of the treatment and therapy system of women addicted in individual countries.

The aim of the study was achieved because the barriers of the addiction treatment system in some countries were shown, and it was also concluded that the change should occur not only at the micro level (of individual treatment facilities), but also at the macro level (of the entire treatment system and health strategies of given countries).

The applied Scoping Reviev qualitative research method is very well operationalized, and its subsequent stages of the analysis are well described. And that is one of the highlights of this review.

The results of the research indicate that WHO should recommend such methods of addiction therapy to many countries of the world, so that they take into account the conclusions described in the study, especially in relation to women. Therefore, the article is not only of scientific and theoretical importance, but also practical and application.

The conclusions of the research are very pessimistic, they even show the secondary victimization of women addicted to drugs, but publicizing the problem will allow for changes in the approach to such patients.

Author Response

Thank you very much for your high appreciation of our work. We wish everyone such high marks for their research initiatives. It is a great joy for us and, above all, the satisfaction of a well-conducted research process and an article that can bring social and scientific benefits.

This manuscript is a resubmission of an earlier submission. The following is a list of the peer review reports and author responses from that submission.

Round 1

Reviewer 1 Report

Authors presented an interesting paper reviewing the barriers and difficulties of women who would attend an addiction service. 

This paper deserves to be published, but it will improve after some review and reorganization of its structure and presentation. For instance, authors present at the begining data on prevalence and epidemiology of drug use in women, and then, it appears again in line 248.

Heading number 3 (Review of selected studies) is difficult to understand (which estudies? why have been selected? How have been selected?).

Also, the reference to transgender and LGTBI populations is interesting, but it deserves a different heading.